# Attention stabilizes the shared gain of V4 populations

**Neil C Rabinowitz[1]\*, Robbe L Goris[1], Marlene Cohen[2], Eero P Simoncelli[1]\***

[1]Center for Neural Science, Howard Hughes Medical Institute, New York University, New York, United States; [2]Department of Neuroscience and Center for the Neural Basis of Cognition, University of Pittsburgh, Pittsburgh, United States

**Abstract** Responses of sensory neurons represent stimulus information, but are also influenced by internal state. For example, when monkeys direct their attention to a visual stimulus, the response gain of specific subsets of neurons in visual cortex changes. Here, we develop a functional model of population activity to investigate the structure of this effect. We fit the model to the spiking activity of bilateral neural populations in area V4, recorded while the animal performed a stimulus discrimination task under spatial attention. The model reveals four separate time-varying shared modulatory signals, the dominant two of which each target task-relevant neurons in one hemisphere. In attention-directed conditions, the associated shared modulatory signal decreases in variance. This finding provides an interpretable and parsimonious explanation for previous observations that attention reduces variability and noise correlations of sensory neurons. Finally, the recovered modulatory signals reflect previous reward, and are predictive of subsequent choice behavior.

\*For correspondence:
neil.rabinowitz@nyu.edu (NCR);
eero.simoncelli@nyu.edu (EPS)

**Competing interests:** The author declares that no competing interests exist.

## Introduction

Sensory information is represented in the activity of populations of neurons, but the responses of individual neurons within these populations are not uniquely determined by external stimuli: repeated presentations of the same stimulus elicit different spike trains. Although some of this variability presumably arises from noise in local circuits, a substantial portion appears to be due to fluctuations in neurons' excitability, i.e. their gain (*Goris et al., 2014*). Since this may result from changes in internal states that modulate sensory responses, such as wakefulness, reward, and expectations, or from changes in top-down signals arriving from other cortical areas, it is likely that a significant component of the gain fluctuations in individual neurons are not "private", but shared across populations (*Ecker et al., 2014*; *Schölvinck et al., 2015*; *Lin et al., 2015*). Shared gain fluctuations could have a crucial impact on sensory computations, depending on their structure (*Moreno-Bote et al., 2014*). Yet this structure remains largely unknown.

Visual attention provides a well-known example of a process that affects neural gain. When monkeys direct their attention to a visual stimulus, the mean firing rates of specific subsets of visual neurons in striate (*McAdams and Maunsell, 1999*; *Herrero et al., 2008*) and extrastriate cortex (*Moran and Desimone, 1985*; *Treue and Maunsell, 1996*; *Treue and Maunsell, 1999*; *Martínez-Trujillo and Treue, 2002*; *Williford and Maunsell, 2006*; *Cohen and Maunsell, 2009*; *Mitchell et al., 2009*) have been found to increase. This motivates the hypothesis that, at the level of the population, attention acts to increase the gain of selected neurons, thereby increasing the signal-to-noise ratio of sensory representations, and hence improving performance on perceptual tasks (*McAdams and Maunsell, 1999*; *Treue and Maunsell, 1996*; *Cohen and Maunsell, 2009*; *Lee and Maunsell, 2009*; *Reynolds and Heeger, 2009*).

**eLife digest** Our brains receive an enormous amount of information from our senses. However, we can't deal with it all at once; the brain must selectively focus on a portion of this information. This process of selective focus is generally called "attention". In the visual system, this is believed to operate as a kind of amplifier that selectively boosts the signals of a particular subset of nerve cells (also known as "neurons").

Rabinowitz et al. built a model to study the activity of large populations of neurons in an area of the visual cortex known as V4. This model made it possible to detect hidden signals that control the attentional boosting of these neurons. Rabinowitz et al. show that when a monkey carries out a visual task, the neurons in V4 are under the influence of a small number of shared amplification signals that fluctuate in strength. These amplification signals selectively affect V4 neurons that process different parts of the visual scene. Furthermore, when the monkey directs their attention to a part of the visual scene, the associated amplifier reduces its fluctuations. This has the side effect of both boosting and stabilizing the responses of the affected V4 neurons, as well as increasing their independence.

Rabinowitz et al.'s findings suggest that when we focus our attention on incoming information, we make the responses of particular neurons larger and reduce unwanted variability to improve the quality of the represented information. The next challenge is to understand what causes these fluctuations in the amplification signals.

This "classical" view of attention has been augmented by recent observations that spatial attention affects more than mean response: in particular, attention reduces normalized measures of spike count variance, as well as stimulus-conditioned spike count correlations (i.e. "noise correlations") across pairs of neurons (*Cohen and Maunsell, 2009*; *Mitchell et al., 2009*; *Herrero et al., 2013*). These measurements (and similar findings in the context of perceptual learning (*Gu et al., 2011*; *Jeanne et al., 2013*), cognitive challenge (*Ruff and Cohen, 2014*), task engagement (*Downer et al., 2015*), or wakefulness (*Poulet and Petersen, 2008*) seem to imply that attention does more than just increase neural gain; for instance, it might change the underlying connectivity of the network. But the origin of these effects is difficult to interpret, as changes in pairwise correlations can arise from changes in direct or indirect couplings between neurons, or changes in common modulatory input (*Goris et al., 2014*; *Ecker et al., 2014*; *Brody, 1999*; *Yatsenko et al., 2015*). In particular, several groups have suggested that variability in the attentional signal itself might contribute to spike count variance and correlation (*Goris et al., 2014*; *Cohen and Maunsell, 2010*; *Harris and Thiele, 2011*; *Ecker et al., 2012*).

Here, we develop a functional model for the population activity that accounts for all the above observations. The model includes stochastic modulatory signals that alter the gains of targeted subsets of neurons. We fit the model to spiking data from populations of ~100 neurons in visual area V4, simultaneously recorded from both hemispheres of a macaque monkey that was performing a change-detection task under directed spatial attention (*Cohen and Maunsell, 2009*). The model is fit without specification of the hemispheric origin of the neurons, or the extent to which they were influenced by the various gain factors. The resulting fitted model reveals that the population was predominantly influenced by two independent shared modulatory signals, each operating primarily within one hemisphere, and each targeting the neurons most relevant for the task. The statistics of each of these signals changed significantly depending on the attentional condition, with each modulator exhibiting a decrease in variance when the monkey was cued to attend to the stimuli in the corresponding hemifield. Together with the (classical) increases in mean response, these changes in the statistics of the shared modulatory signals account for the previously-reported decrease in neural variability and noise correlations. Finally, we show that the inferred modulatory signals are correlated with the monkey's behavioral performance on each trial, and are influenced by the reward received on the previous trial. The structure and statistics of shared modulatory fluctuations thus provide a parsimonious account of attentional effects on population coding and behavior. A preliminary account of these results was presented in *Rabinowitz et al. (2015a)*.

## Results

We analyzed spiking data from populations of single- and multi-units recorded in visual area V4 of two macaques while they performed a visual detection task. The experiment and data are described in detail in *Cohen and Maunsell (2009)*. We briefly describe the relevant details here.

Monkeys fixated on a central mark while two oriented grating patches were flashed concurrently on opposite sides of the fixation point. These stimuli (the "standards") were shown for 200 ms, with a delay of 200–400 ms between presentations. After a variable-length sequence of repeated standards, the orientation of one of the two gratings (the "target") was changed. For a small fraction of trials ("catch trials"), no target appeared. The monkey had to detect the target, and to indicate this with a visual saccade to its location. Trials were grouped in blocks of 125, in which the target had an 80% probability of occurring on one side. The identity of this "cued side" was indicated to the monkey through a set of 10 instruction trials preceding each block.

While the monkey performed this task, the spiking responses of populations of neurons in visual area V4 were collected from two microelectrode arrays, one per hemisphere. A total of 50–130 units were concurrently recorded from the two hemispheres per day. Over 36 recording days, 3004 units were collected and deemed suitable for further analysis. We analyzed spiking responses from these populations to a total of ~$10^5$ presentations of the standard (i.e. non-target) stimuli. We excluded responses to the first standard stimulus in each sequence (since those responses exhibit atypical onset transients), and to any stimulus in which the monkey failed to maintain fixation. We examined the total spike count for each 200 ms stimulus period, offsetting the response window by 60 ms to account for stimulus-response latency (as in *Cohen and Maunsell, 2009*).

### Fitting the shared modulator model

We used a computational model to explore the structure of neural population activity in the presence of spatial attention (*Figure 1*). We describe the stimulus-driven instantaneous firing rate of each neuron $n$ over time as $f_n(s(t))$, where the function $f_n(\cdot)$ describes the mapping from stimulus, $s(t)$, to that neuron's firing rate. In addition, we allow each neuron's gain to be affected by three signals: (1) the current attentional cue, $c(t)$, (a binary signal); (2) a slowly-varying global drift, $d(t)$; and (3) a set of shared time-varying modulators, $m_k(t)$. We assume that there are $K$ such modulatory signals, indexed by $k$. The degree to which each of these three signals affects the gain of each neuron is specified by a set of coupling weights, $\{u_n, v_n, w_{n,k}\}$, and the net instantaneous firing rate of the neuron is written as:

$$r_n(t) = f_n(s(t)) \cdot \exp\left[ u_n \cdot c(t) + v_n \cdot d(t) + \sum_{k=1}^{K} w_{n,k} \cdot m_k(t) \right] \qquad (1)$$

The exponential acts to convert the weighted sum of the three signals, which may take on positive or negative values, into a product of three positive-valued modulatory quantities.

Importantly, only the stimulus $s(t)$ and cue signal $c(t)$ are known. The drift, $d(t)$, and modulators, $m_k(t)$, as well as all of the coupling weights must be fit to the experimental data. To accomplish this, we assumed that the firing rate of each neuron, $r_n(t)$, was constant over the duration of each stimulus presentation (200 ms), and that the observed spike counts arose from a Poisson distribution with that rate. We then fit this probabilistic model to the spike count data by maximizing the posterior probability of the model parameters (see Materials and methods). We describe each of the component signals in turn.

The stimulus and cue signals encapsulate the external factors that are set by the experimenter and available to the monkey. Since we only analyzed responses to the standard stimuli— which were identical for all trials of a given day—the stimulus-dependent drive in this experiment, $f_n(s(t))$, is captured by a single mean firing rate per neuron. For clarity of presentation, we assume that this is the firing rate when the monkey was cued away from this stimulus (i.e. to the opposite side). The response of each neuron is then affected by the cue signal to a different degree, which is captured by an additional free parameter per neuron (the "coupling weight" to the cue signal), $u_n$. Together, these two factors comprise the "classical" model of attention, wherein the attentional cue results in a change in the mean firing rate of each neuron.

The second modulatory signal, $d(t)$, is motivated by recent reports of coordinated, global slow fluctuations in neural response. These have been hypothesized to arise from fluctuations in global

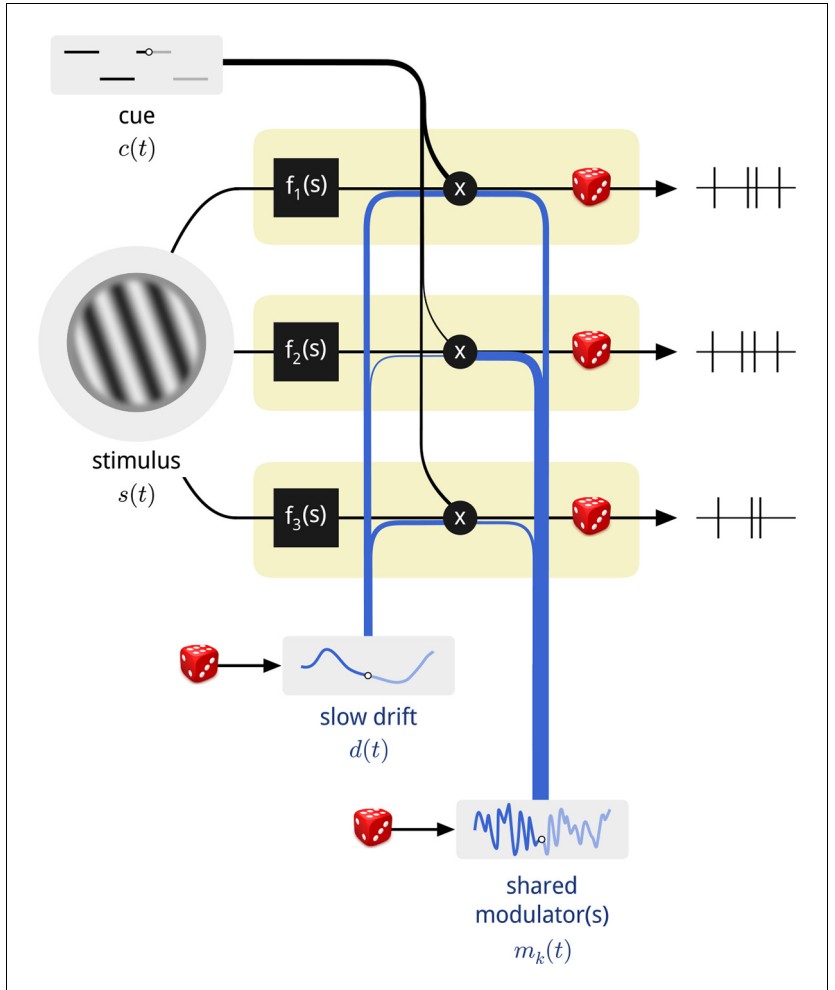

**Figure 1.** Diagram of the modulated population model. Shown are three neurons (yellow boxes), each with a firing rate that is a function of the stimulus multiplied by three time-varying gain signals: the binary attentional cue; a slow global drift; and a set of shared modulators. The influence of each of these signals on each neuron is determined by a coupling weight, indicated by the thickness of the blue and black lines. Only one shared modulator is shown in this schematic, but the model allows for more, each with its own coupling weights. The spike counts of each neuron are conditionally Poisson, given the firing rate.

The following figure supplements are available for figure 1:

**Figure supplement 1.** Example slow drifts in spike counts of four simultaneously-recorded units (two from each hemisphere), taken from a recorded population of 77 units.

**Figure supplement 2.** Signatures of modulatory (multiplicative) effects in the neural responses.

state variables such as arousal, with different neurons potentially affected by these signals to different degrees (*Goris et al., 2014*; *Ecker et al., 2014*; *Schölvinck et al., 2015*; *Okun et al., 2015*; *McGinley et al., 2015*). The V4 data analyzed here also exhibit slow global drifts in neural gain in each recording, with typical time constants on the order of minutes to tens of minutes (*Figure 1—figure supplement 1*). We therefore allowed a single global slow signal to affect the population. Its timescale and time course were inferred from the population responses, as well as its coupling weights to each neuron (see Materials and methods). While this substantially improved model predictions (see below), global state fluctuations have been described elsewhere and are not the main focus of this work. Importantly, the inclusion or exclusion of this global drift signal (even allowing more than one) does not qualitatively change the results or interpretation presented here.

Our primary interest was in the $K$ fast modulatory signals, $m_k(t)$, which we introduced to explain any shared structure in the population activity that remains after the structure of the task and global slow drifts have been accounted for. We assumed that these consisted of a small number of independent sources. The number of sources was itself a free parameter of the model, which we explore below.

To evaluate the model fits, we held aside 20% of the spike counts from the full set of stimulus presentations, randomly chosen across units and time. The accuracy of model predictions for these held-out data improved substantially with the inclusion of each model component (*Figure 2a*). Since all scores are *cross-validated*, these improvements indicate that the models are explaining structure in the data rather than overfitting to random response fluctuations. The figure shows that the shared modulators were twice as important as the experimentally-controlled attentional cue signal in making predictions on held-out data. This improvement is remarkable given that the stimulus presented was identical on every trial: these model components are capturing a substantial portion of the trial-to-trial variability in the population responses.

## Dimensionality and connectivity of shared modulation

We next asked how many independent shared modulators were needed to explain the recorded responses. In principle, the activity of the recorded populations could be influenced by a very large number of shared modulators, each with its own unique connectivity and temporal patterns. We found that the spike count predictions on held-out data improved substantially with the inclusion of two modulators, and showed a modest additional increase for up to four modulators. Including additional modulators beyond these did not improve predictions.

To verify that this outcome did not reflect insufficiency of the dataset in constraining the model, we generated synthetic datasets by simulating responses of the model with different numbers of modulators, each of equal strength (*Figure 2—figure supplement 1*). These synthetic datasets had the same number of neurons and trials as the true data, and we scaled the magnitude of the simulated modulators' fluctuations to produce the same average noise correlations as found in the true data. We found that we could accurately recover up to 8 independent modulators from these synthetic datasets – more than the number recovered from the true data.

We next asked how the estimated modulators, and their associated weights, were structured with respect to the neural populations and the task. These structures are most easily visualized for the two-modulator model, and we therefore restrict our analysis to this case for the remainder of this article. Results for three- and four-modulator models are given in *Figure 2—figure supplement 2*.

Several striking patterns emerge in the fitted model. First, although the model was not given any information regarding anatomical location or connectivity of the neurons, the estimated coupling weights for each modulator clearly identify the hemisphere in which the corresponding neurons reside (*Figure 2b–c*). In a given recording, each modulator had largely positive weights for neurons in one hemisphere (indicating that these neurons' gains were being co-modulated by this signal), and small weights for neurons in the opposite hemisphere. Thus, the weight structure of the estimated model suggests that the two hemispheric V4 subpopulations are modulated by two independent signals. Based on this observation, we examined a restricted model, in which we explicitly enforce the hemispheric assignment of each modulator (from here on, referred to as the LHS and RHS modulators), by setting the weights for neurons in the opposite hemisphere to zero. This enforced assignment reduces the number of weight parameters by a factor of two, and results in a modest improvement in the quality of model predictions on held-out data, reaching the level of the 4-modulator model (*Figure 2a*, grey square). For the remainder of this article, we retain this enforced assignment of the modulators to the hemispheres.

In addition to the distinct anatomical connectivity of the two modulators, we found that they also exhibited specific *functional* connectivity. We characterized the task-relevance of each neuron in terms of its ability to discriminate the standard and target stimuli in the cue-away (inattentive) condition. Specifically, for each neuron, we computed the difference in mean spike count for the two stimuli, relative to the standard deviation (known in the perceptual psychology literature as $d'$). Comparison of these values to the modulator coupling weights indicates that the modulators preferentially targeted the most task-informative neurons ($r = 0.42$; *Figure 2d*). One might suspect that some portion of this effect is simply due to firing rate—units with higher mean firing rates typically had stronger coupling weights to the modulators, paralleling previous observations (*Cohen and*

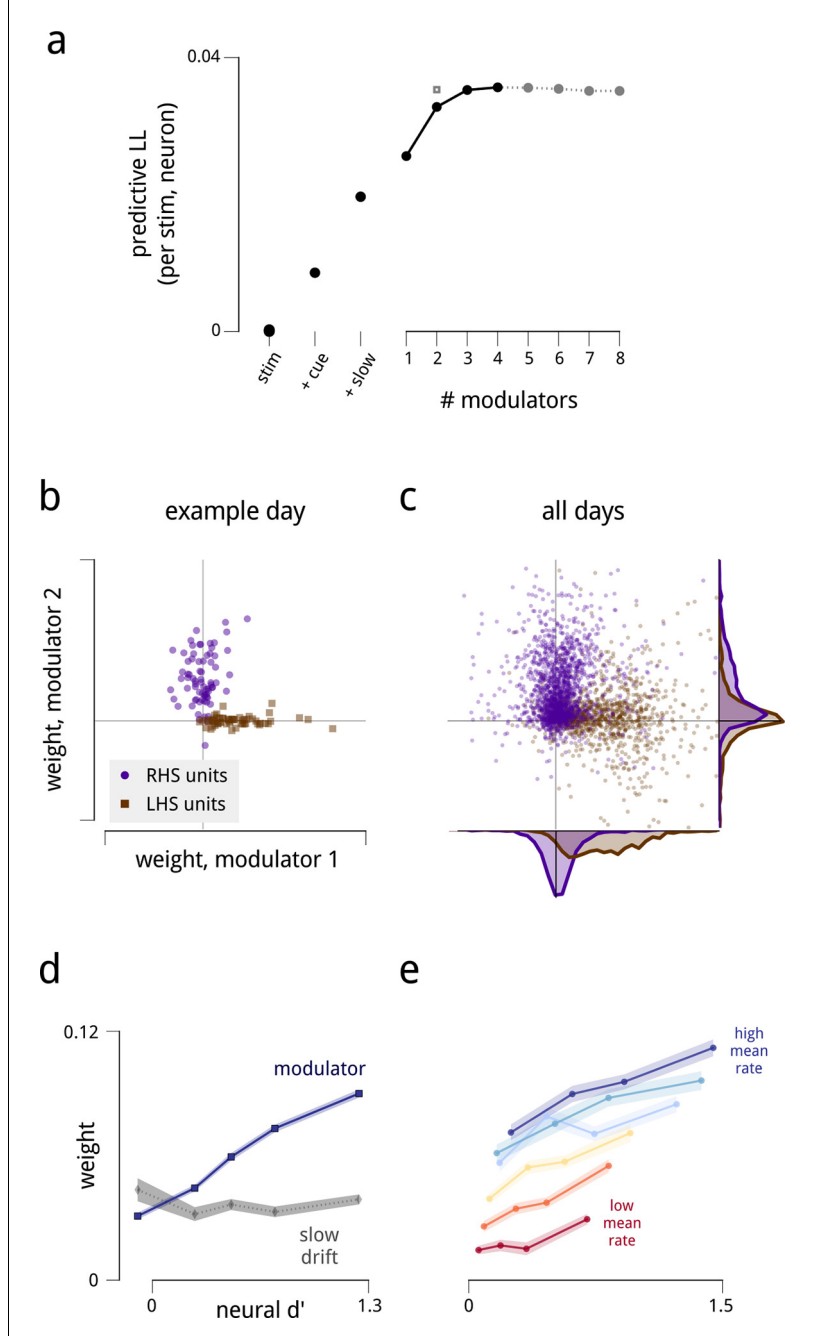

**Figure 2.** The fitted model explains the observed spiking responses, with estimated modulators that are both anatomically and functionally targeted. (a) Performance comparison of various submodels, measured as log-likelihood (LL) of predictions on held-out data. Values are expressed relative to performance of a stimulus-drive-only model (leftmost point), and increase as each model component (cue, slow drift, and different numbers of shared modulators) is incorporated. The grey square shows the predictive LL for a two-modulator model, with each modulator constrained to affect only one hemisphere (i.e. with coupling weights set to zero for neurons in the other hemisphere). This restricted model is used for all results from *Figure 2d* onwards, excepting the fine temporal analysis of *Figure 6c*. (b) Modulators are anatomically selective. Inferred coupling weights for a two-modulator model, fit to a population of units recorded on one day. Each point corresponds to one unit. As the model does not uniquely define the coordinate system (i.e. there is an equivalent model for any rotation of the coordinate system), we align the mean weight for LHS units to lie along the positive x-axis (see Materials and methods). (c) Distribution of inferred coupling weights aggregated over all recording days indicates that each shared modulator provides input primarily to cells in one hemisphere. (d) Hemispheric modulators are

*Figure 2 continued on next page*

*Figure 2 continued*

functionally selective. Units which are better able to discriminate standard and target stimuli in the cue-away condition have larger coupling weights (blue line). Discriminability is estimated as the difference in mean spike count between standard and target stimuli, divided by the square root of their average variance ($d'$). Values are averaged over units recorded on all days, subdivided into five groups based on their coupling weights. Shaded area denotes ±1 standard error. Pearson correlation over all units is $r = 0.42$. This relationship is not seen for the weights that couple neurons to the slow global drift signal (gray line, Pearson correlation $r = 0.00$). The relationship between $d'$ and cue weight is significant, but weaker than for modulator weight ($r = 0.24$); this is not shown here as the cue weights are differently scaled. (e) Same as in (d), but with units subdivided into subgroups according to mean firing rate. Each line represents a subpopulation of ~500 units with similar firing rates (from red to blue: 0–7; 7–12; 12–17; 17–25; 25–35; 35–107 spikes/s). Within each group, the Pearson correlations between $d'$ and coupling weight are between 0.2–0.3, but the correlations between mean rate and coupling weight are weak or negligible.

The following figure supplements are available for figure 2:

**Figure supplement 1.** The dataset is sufficient to support the estimation of up to 8 shared modulators.

**Figure supplement 2.** The structure of the modulators in higher-dimensional modulator models.

**Figure supplement 3.** The modulators' anatomical, functional, and attentional structure manifests primarily within the dominant two dimensions of modulation.

**Figure supplement 4.** Units with higher mean firing rates typically had stronger coupling to their respective population modulator ($r^2 = 0.21$).

*Maunsell, 2009*; *Mitchell et al., 2009*; *de la Rocha et al., 2007*; *Ecker et al., 2010*) (*Figure 2—figure supplement 4*). But the relationship between task informativeness and modulator coupling remains robust even when conditioned on firing rate (*Figure 2e*). Finally, this correlation with functional specificity was weaker for the coupling weights to the cue signal ($r = 0.24$), and entirely absent from the coupling weights to the slow global drift signal ($r < 0.01$; *Figure 2d*).

## The effect of attention on shared gain

The model thus far reveals the action of two structured modulatory signals, each providing input to task-informative V4 neurons in one of the two hemispheres. Given that these signals were recovered from population activity during a cued attentional task, we next ask whether they exhibit systematic changes across differently cued blocks.

The model recovers estimates of the slow drift signal and each hemisphere's modulator for every stimulus presentation (*Figure 3a*). Here, a clear pattern is evident: when the monkey was cued to attend to one visual hemifield, the shared modulator of the corresponding (contralateral) V4 population had a smaller variance (*Figure 3b*). The modulators' mean values were unchanged across blocks, as any changes in mean rate are captured by the coupling to the cue signal: 75% of neurons had a positive coupling weight to the cue signal, capturing an increase in their firing rate under cued attention. Thus, in general, attention both *increases* and *stabilizes* the time-varying gain of the corresponding neural population.

These two changes in shared gain, in turn, provide a simple explanation for the observed changes in the statistics of individual and paired neural spike counts. Consider first a simulation of two conditionally-Poisson neurons. In the classical model of attention, these neurons' gains increase when the cue is directed to the appropriate hemifield. This produces two major effects on each neuron's marginal spike count statistics (*Figure 4a*): the mean increases, and the variance goes up as well (due to the Poisson mean-variance relationship). The Fano factors (ratio of variance to mean) remain unchanged.

Next, consider what happens if these simulated neurons are coupled to the shared modulator (*Figure 4b*). A decrease in the variance of this shared signal leads to a reduction in the spike count variance of each neuron, without a large change in their mean firing rates (*Goris et al., 2014*). Consequently, their respective Fano factors decrease. Moreover, since both neurons are coupled to the

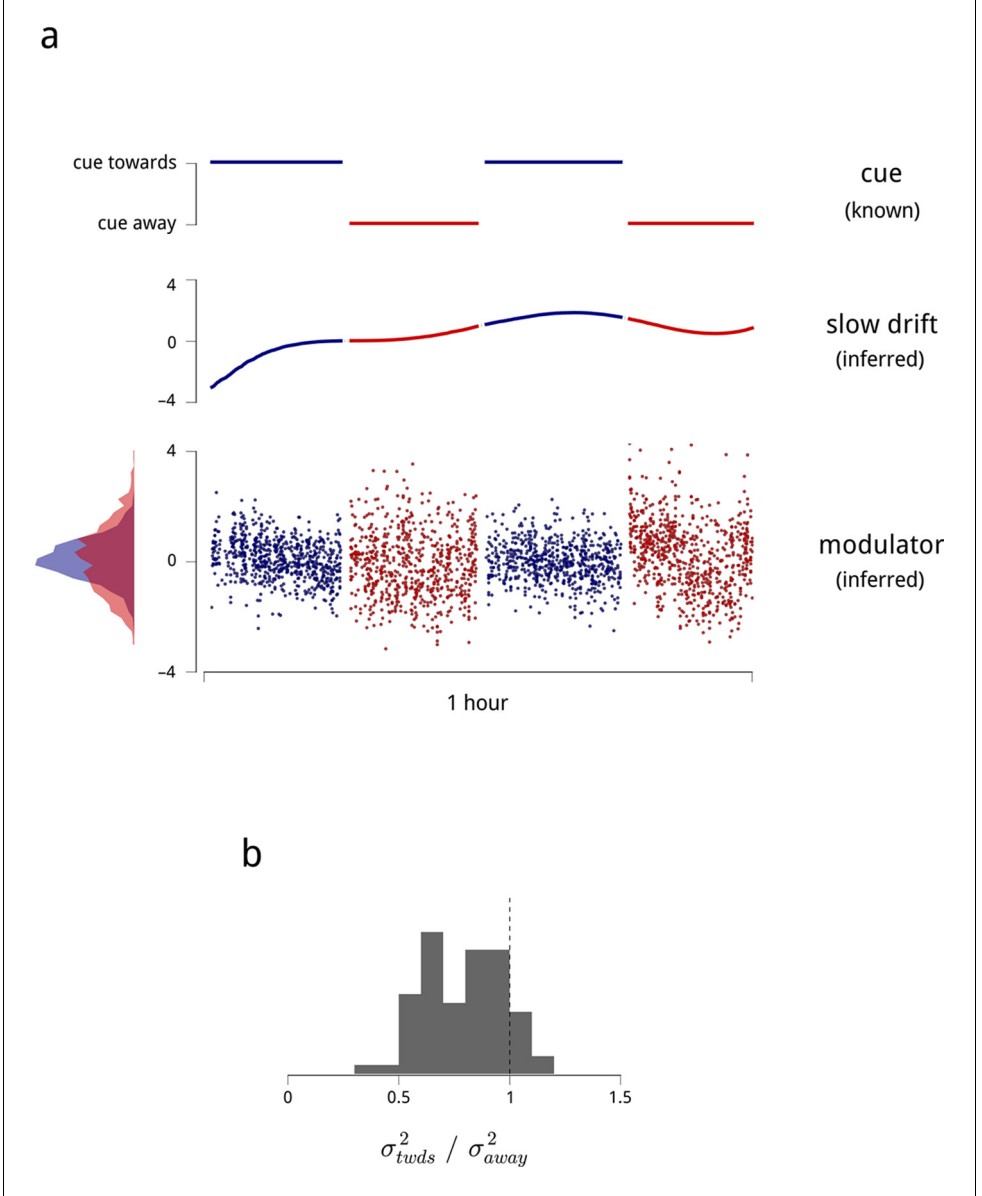

**Figure 3.** Time-varying model signals that determine the gain of units in one V4 hemisphere. (**a**) Example values of the cue signal (imposed by experiment), the slow drift (inferred), and a single hemispheric modulator (inferred) across stimulus presentations for one day and hemisphere. In the model, the gain of each neuron is obtained by exponentiating a weighted sum of these three signals (see *Equation (1)*). Histogram in the bottom left shows the distribution of modulator values when the monkey was cued towards the contralateral side (blue), and away from it (red). (**b**) Modulator variance decreases under cued attention. Histogram shows the ratio of modulator variances estimated in the two cue conditions. Averaged across days and hemispheres, cued attention reduces modulator variance by 23%.

same modulatory signal, the decrease in modulator variance also causes a decrease in the spike-count correlation of the pair.

The data recorded in the experiment exhibit a combination of these effects. An example is shown in *Figure 4c*. Under cued attention, the variance of this V4 hemisphere's shared modulator decreases. When we consider two neurons within this population with strong coupling weights to both the cue signals and the modulators, their marginal and joint spike count statistics are changed

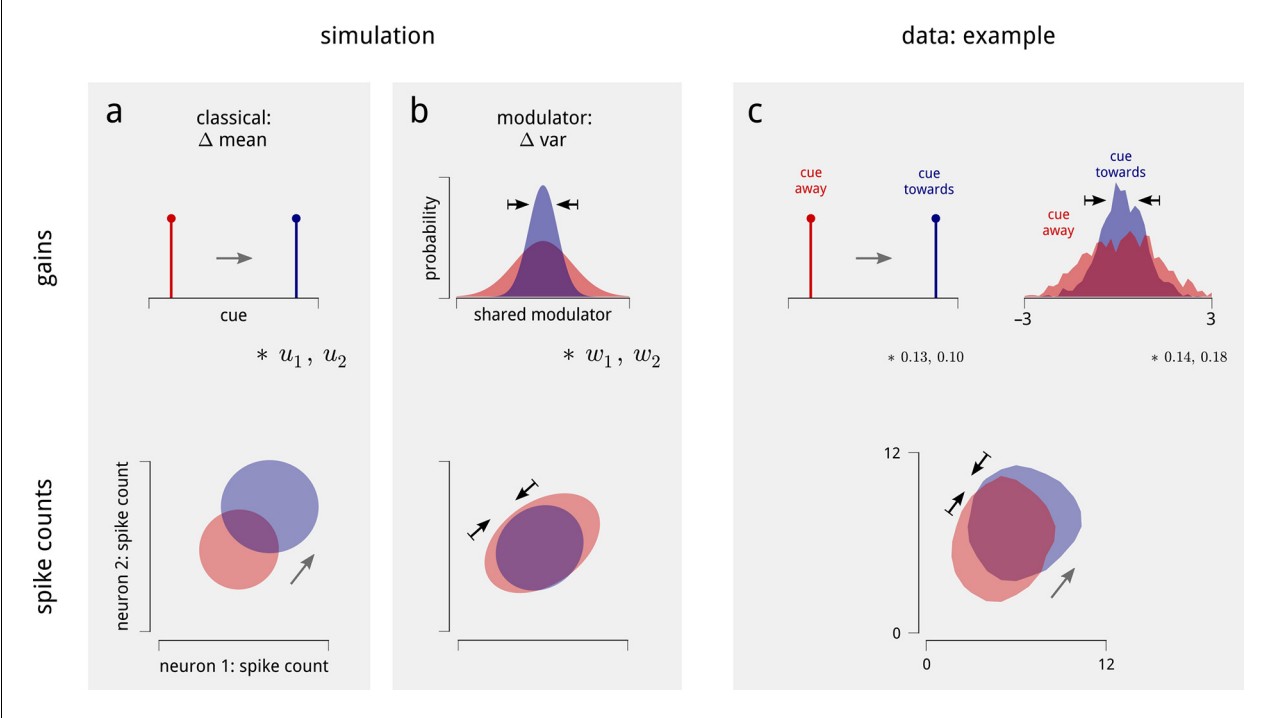

**Figure 4.** Changes in the statistics of the inferred modulator under cued attention explain the observed changes in spike count statistics. (**a**) The classical model of attention. Simulation of two neurons with positive coupling weights, $u_1$ and $u_2$, to the cue signal. When the cue is directed to the corresponding spatial location (top), both the mean and variance of the simulated neurons' spike counts increase (bottom). Shaded areas demarcate analytic iso-density contours, i.e. the shape of the joint spike count distributions. (**b**) The effect of the shared modulator. Simulation of two simulated neurons with positive coupling weights, $w_1$ and $w_2$, to a shared modulator. A decrease in modulator variance leads to a decrease in both the variance and correlation of spike counts (bottom). (**c**) Effects on an example pair of units within the same hemisphere, on one day of recording. The cue increases the gain of both cells (numbers indicate cue coupling weights), and the inferred modulator exhibits a decreased variance in cued trials (again numbers indicate coupling weights; top). The spiking responses of the cells exhibit a combination of the effects simulated in (**a**) in (**b**): increased mean, decreased Fano factor, and decreased correlation (bottom; means from 7.0 to 8.1 and 6.0 to 7.2 spikes/stim respectively; Fano factors from 1.9 to 1.6 and 1.7 to 1.6 respectively; correlation from 0.19 to 0.10). The shaded areas demarcate smoothed iso-density contours estimated from the data.

by attention as predicted by the simulation: their mean firing rates go up, their Fano factors go down, and their noise correlations decrease.

The behavior shown in *Figure 4* crucially depends on the two neurons being coupled to the shared modulator. The model thus makes a very specific prediction: the magnitudes of the Fano factors and noise correlations should increase with the magnitude of the weights with which the neurons are coupled to the hemispheric modulators. We find that this prediction is clearly borne out by the data (*Figure 5a*), and is robust when controlled for firing rate (*Figure 5—figure supplement 1a*). In addition, since the modulator variance decreases under cued attention, the model predicts that those neurons that are more strongly coupled should exhibit a larger attention-induced reduction in these measures. This effect is also apparent in the data (*Figure 5a*). Ultimately, the model accounts for the majority (but not all) of the attention-induced changes in Fano factor and noise correlation (*Figure 5b*). Thus, our population-level model accounts not only for the single neuron statistics and pairwise correlations, but also for the diversity of effects seen across the population.

The model does not assume any relationship between the coupling weights for cue and shared modulators. We did observe that neurons which were most strongly coupled to the modulators were also most strongly coupled to the cue signal (*Figure 5c*; even when controlled for firing rate *Figure 5—figure supplement 1b*). But overall, the correlation between the coupling weights to the cue signal and the coupling weights to the hemispheric modulator was modest (Spearman $\rho = 0.26$). We fit a restricted model in which these two signals had identical coupling weights, and found that the predictive (log-likelihood) performance was reduced by 20%. Moreover, while the modulator weights

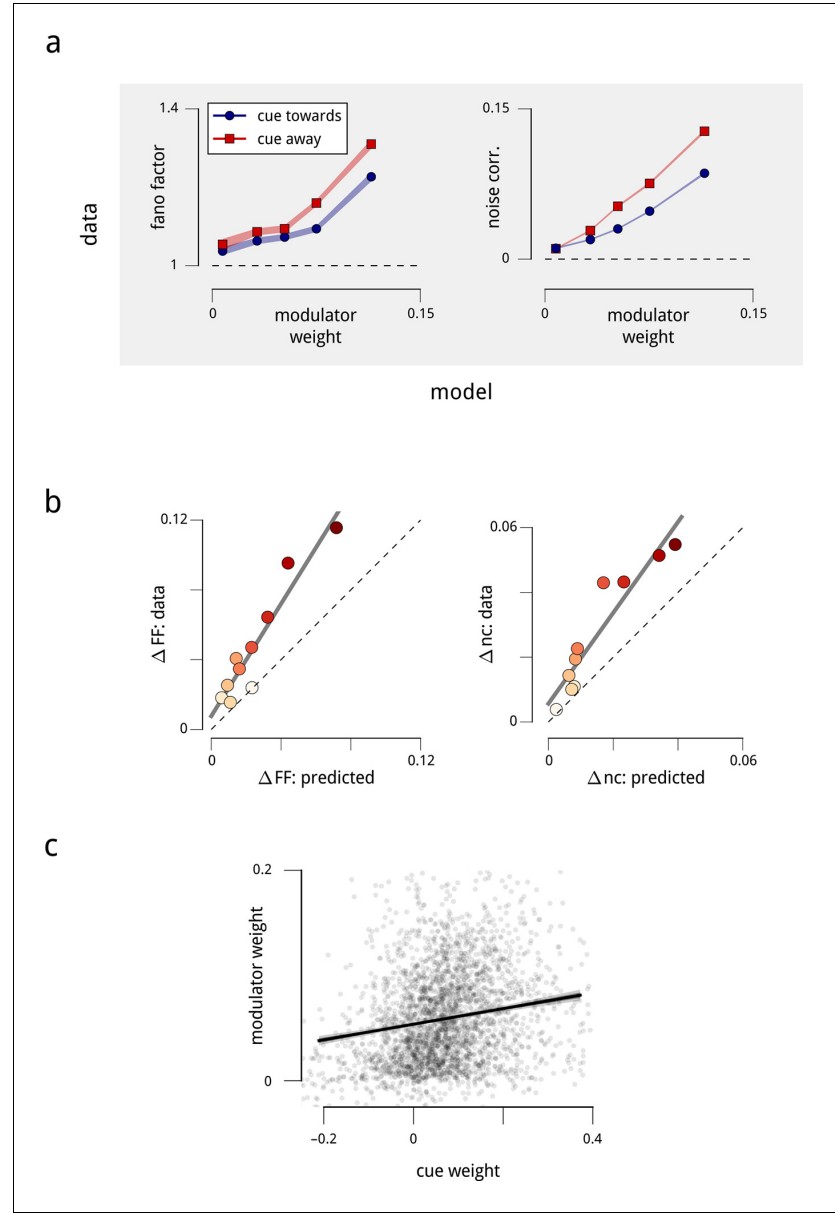

**Figure 5.** Attention-induced changes in neural response statistics are larger for neurons that are more strongly coupled to the shared modulator. (a) Observed Fano factor and noise correlations, as a function of model coupling weight. Units from all days are divided into five groups, based on their fitted coupling weight to their respective population modulator (model-based quantities), as in *Figure 2d*. Points indicate the average Fano factors and noise correlations (model-free quantities) within each group, when attention was cued towards the associated visual hemifield (blue) and away from it (red). Shaded area denotes ± 1 standard error. Unitwise Spearman correlations: $\rho$ = 0.31/0.44/0.40/0.51 (fano cue twds/fano cue away/ncorr cue twds/ncorr cue away). (b) Comparison of model-predicted vs. measured decrease in Fano factor and noise correlation. Units are divided into ten groups, based on coupling weights (darker points indicate larger weight). The model accounts for 62% of the cue-induced reduction in Fano factor, and 71% of the reduction in noise correlation. (c) Comparison of cue weights and modulator weights. Units that are strongly coupled to the cue signal are typically strongly coupled to the modulator signal, though the relationship is only partial (unitwise Spearman correlation: $\rho$ = 0.26). These results are robust when controlled for firing rate (*Figure 5—figure supplement 1*).

The following figure supplement is available for figure 5:

**Figure supplement 1.** Differences in coupling weight explain the observed statistics of single and pairwise firing rates, even when controlled for mean firing rate.

were predictive of the Fano factor and noise correlation effects (as shown in *Figure 5a*; correlations $\rho > 0.3$), the cue weights were much less so ($\rho < 0.1$). Thus, although the effects of the cue and of the inferred hemispheric modulator overlap, they are not identical, suggesting that they arise (at least in part) from distinct sources.

We made three additional observations regarding the shared modulators. First, the two hemispheres' time-varying modulator values were almost completely uncorrelated (*Figure 6a*; $r = 0.03$). The very small positive correlation might result from the influence of a global signal spanning both hemispheres. Second, the modulators exhibited correlations over successive stimulus presentations (*Figure 6b*), indicating that gain fluctuations can persist over time scales of seconds. Third, while the data were insufficient to infer the value of the modulators at a sub-trial resolution (i.e. at time scales shorter than 200 ms) for individual trials, we were able to estimate their average sub-trial time course. We found that the shared modulators had only weak effects during response onsets, and exerted their influence on neural gain primarily during the sustained response period (*Figure 6c*). This property of the modulators is consistent with previous reports that attentional effects on firing rates (and Fano factors) are greater during sustained periods (*Cohen and Maunsell, 2009*; *Mitchell et al., 2009*, and with more general observations that network behavior is less affected by context or state during onset transients (*Churchland, 2010*).

Finally, we note that the relationships between attention and modulator statistics were not only evident in the hemispheric-modulator model, but were also present in models with more modulators (*Figure 2—figure supplement 2*). Nevertheless, these additional modulators did not reveal any additional anatomical or functional specificity, or attentional dependencies (*Figure 2—figure supplement 3*).

## Relationship of modulation to behavior

We have thus far used the model to expose a set of internal modulatory signals that selectively affect the gains of neurons in the population. We wondered whether these modulatory signals had any effect on behavior. For clarity in describing these results, we use the terms "cued" and "opposite" rather than "left" and "right" to describe the visual hemifields, V4 populations, modulators, and targets. For example, if the monkey was cued to the left during a block, then the left hemifield is cued, the monkey's right (i.e. contralateral) V4 is cued, and the right V4 population's modulator is the cued modulator. If, during a trial in this block, a target is presented on the right, we refer to it as an opposite target.

First, we asked whether the shared modulators had any influence on the monkey's trial-by-trial performance. We found that the values of both the cued and the opposite modulators, during presentations of the standard stimuli, were predictive of whether the monkey would detect the upcoming target stimulus. As may be expected, changes in each hemispheric modulator predicted performance changes for their associated targets: an increase in the cued modulator preceded an increased detection probability for cued targets; while an increase in the opposite modulator preceded higher detection probabilities for opposite targets (top left and bottom right of *Figure 7a*; see *Figure 7—figure supplement 1a* for full psychometrics). Suprisingly, this effect is substantially stronger for the opposite side. This result is similar to a previous study on this dataset aimed at directly decoding the trial-by-trial attentional state of the animal (*Cohen and Maunsell, 2010*), though the two results rely on different readouts of the population activity.

In addition to this modulator-driven improvement in detection probabilities, we also uncovered a striking deficit in performance on the opposite side. Specifically, an increase in one side's modulator predicted a decrease in detection probability on the other side (top right and bottom left of *Figure 7a*). This is not due to anti-correlation of the two modulators (they are nearly uncorrelated; *Figure 6a*). Rather, this pattern likely reflects the competitive structure of the task, which requires the animal to make comparisons between the represented stimuli in both hemispheres. The influence of V4 activity on detection probability in this task thus cannot be fully explained by a 1D "axis of attention" (*Cohen and Maunsell, 2010*), but depends on the joint (2D) gain modulation across the two hemispheres.

Second, we examined the relationship of the modulator values to the outcome of the previous trial. It is well known that both humans and animals show serial dependence in behavioral tasks: our choices in a task are biased by the percepts, actions and outcomes of previous experience (*Senders and Sowards, 1952*; *Green, 1964*; *Lau and Glimcher, 2005*; *Busse et al., 2011*;

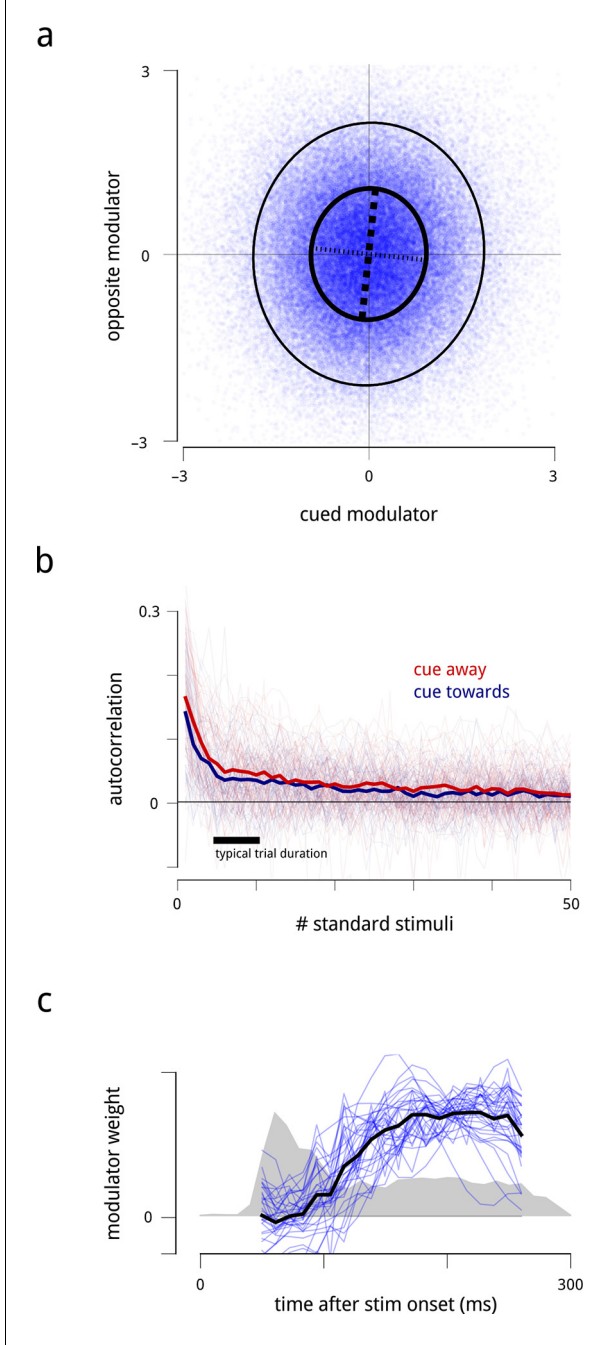

**Figure 6.** Statistical properties of the hemispheric modulators. (a) Joint statistics of the two hemispheric modulators. Blue points: simultaneous values of the two modulators aggregated over all days. Thick black ellipse: iso-density contour at one standard deviation of the Gaussian density matching the empirical covariance. Thinner black ellipse: two standard deviations. Dashed lines: principal axes (eigenvectors) of this covariance, with the thicker dashed line indicating the axis with the larger eigenvalue. The vertical elongation of the ellipse shows that the variance of the modulator for the cued side is smaller than the variance of the modulator for the opposite side. The slight clockwise orientation shows that the two modulators have a very small positive correlation ($r = 0.03$, $p$ negligible). (b) Autocorrelation of modulators across successive stimulus conditions. Individual lines show the within-block autocorrelation of each estimated modulator; the thick lines shows the average across days and hemispheres. For simplicity of presentation, the targets and the gaps between trials have been ignored. The time constant of this process is on the order of several seconds. (c) Average time course of shared modulation within each stimulus presentation. We extended the population response model by allowing the value of the modulator to change over the course of each stimulus presentation. Given limitations of the data at fine temporal resolutions,

*Figure 6 continued on next page*

*Figure 6 continued*
we assumed that the temporal evolution of the modulator within each stimulus presentation followed some stereotyped pattern (up to a scale factor that could change from one stimulus presentation to the next; see Materials and methods). Fine blue lines: modulators' (normalized) temporal structure extracted for each recording day. Heavy black line: average across days. Grey shaded area: normalized peri-stimulus time histogram (arbitrary units) of spiking responses during presentations of the standard stimuli, averaged across all units, days, and cue conditions, with zero denoting spontaneous rate. Shared modulation predominantly occurs during the sustained period and is nearly absent during the onset transient.

*Fischer and Whitney, 2014*). The monkeys in this task also exhibit a serial dependence: after receiving a reward for correctly identifying a cued target, they were more likely to score a hit for a cued target, and a miss for an opposite target, on the next trial. Conversely, after receiving a reward for correctly identifying an opposite target, on the next trial they are substantially more likely to score a hit for an opposite target, and a miss for a cued target (*Figure 7b*).

This sequential bias is explicitly captured in the modulatory signals controlling the gains of the V4 populations. A hit (and thus, a reward) on a cued target biased the two modulators on the subsequent trial to shift in favor of the cued population, while a hit on the opposite target biased the two modulators on the subsequent trial to shift in favor of the opposite population (*Figure 7c*). Again, the effect is more substantial for the modulator opposite the cue. This effect is consistent (in sign) with the biasing effects of previous reward on the animal's behavior. However, we note that only 5–10% of the total serial dependence can be explained through the V4 modulators (*Figure 7—figure supplement 1b*). We conclude that reward must also induce additional biases in the activity of downstream neurons involved in the animals' decisions.

In summary, we found that the inferred signals that modulate neural gain in V4 populations are intimately connected with the monkey's previous and subsequent behaviors.

## Discussion

We have used a statistical model to examine the shared gain fluctuations of a large population of V4 neurons, under the influence of cued spatial attention (*Figure 1*). When fit to measured population activity, the model reveals the hidden influence of 2–4 shared modulatory signals (*Figure 2a*). The two most significant modulators exhibit pronounced anatomical, functional and attentional structure: they each target neurons in one hemisphere (*Figure 2b–c*); they are most strongly coupled to the most task-relevant neurons (*Figure 2d–e*); and their temporal statistics change under cued attention (*Figure 3*). Specifically, when attention is directed to one hemifield, the modulatory signal associated with the corresponding V4 hemisphere decreases in variance, thus stabilizing the shared gain fluctuations within that subpopulation. This, in turn, provides an interpretable and parsimonious account of the observed effects of attention on V4 responses: a reduction in variability (Fano factor) and pairwise spike count correlation (*Figure 4*). The estimated coupling strengths to the modulatory signals predicts the degree to which individual neurons exhibit these effects (*Figure 5*). The two hemispheric modulators are nearly uncorrelated with each other, but show temporal structure on both fine (<100 ms) and coarse (>1 − 10s) time scales (*Figure 6*). Finally, the inferred modulatory signals are correlated with both the rewards received on a previous trial, and successful responses in a current trial (*Figure 7*).

### Relationship to other results

Our findings are consistent with previous reports that, under attention, the activity of single V4 neurons changes in relation to aggregate activity. Specifically, attention reduces the correlation between spiking and concurrent slow fluctuations in local field potential (LFP) (*Fries et al., 2001*), which themselves are reduced in power (*Fries et al., 2001*; *Siegel et al., 2008*). These behaviors are predicted by our model as a simple consequence of the attention-driven changes in modulator variance. Since attention reduces the variance of the shared modulator, it would also be expected to reduce the variance of summed activity across the population, such as that measured in LFPs. In addition, since

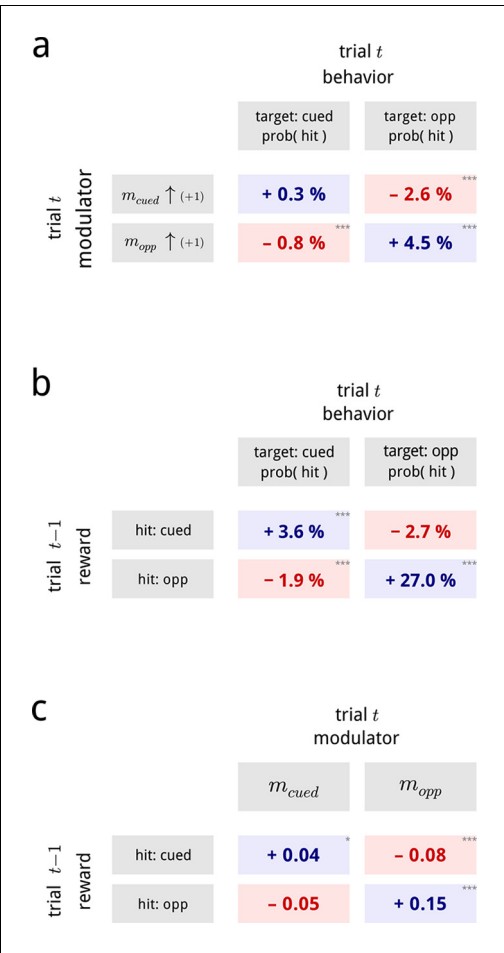

**Figure 7.** Inferred modulatory signals are predictive of behavioral performance, and are influenced by previous reward. (**a**) Average effect of modulator values on subsequent behavioral performance, averaged across all days and difficulty levels. Values show the average change in hit probability for targets on the cued side (left column) and the opposite side (right column) following a unit increase in the cued (top row) and opposite (bottom row) modulators. $*p < 0.05$, $**p < 0.01$, $***p < 0.001$. Full psychometric curves are shown in *Figure 7—figure supplement 1a*. (**b**) Average effects of previous trial reward on current trial performance. Note that this is a direct comparison of the behavioral data, and does not involve the modulator model. (**c**) Average effects of previous trial reward on the value of the two hemispheric shared modulators.

The following figure supplement is available for figure 7:

**Figure supplement 1.** Relationship between modulators and behavior: additional details.

attention reduces the proportion of neural variance due to shared modulation, the correlation between single neuron spike counts and the aggregate LFP activity would also be reduced (*Goris et al., 2014*).

A primary assumption of our model is that the patterns of shared response variability arise from stochastic signals that modulate the gain of sensory neurons. The multiplicative nature of this inter-action is broadly consistent with the patterns of response variability seen throughout visual cortex, which show signatures of multiplicative, rather than additive noise (*Goris et al., 2014*). Nevertheless, other modeling efforts have found evidence for shared additive noise (*Lin et al., 2015*), and we can-not fully rule out the possibility that additive noise also contributes to the variability in our data. A strong test of the multiplicative assumption requires analysis of neural responses over a range of stimulus drive levels. In the dataset analyzed here, reliable responses to only one stimulus per neuron are available for analysis (target responses have not been included, since they are corrupted by co-

occurring saccades). Nevertheless, the modest reduction in firing rates arising from adaptation to repeated presentations of the standard stimuli provides some opportunity to examine this question. We find that the patterns of response variance along this axis are indeed consistent with the multiplicative hypothesis (*Figure 1—figure supplement 2*). A more definitive comparison of additive vs. multiplicative interactions could be achieved with more extensive experimental manipulations of stimulus parameters.

One of us recently reported that pairs of neurons within the same hemisphere whose responses provide evidence for opposite behavioral choices can exhibit increased noise correlations under spatial attention (*Ruff and Cohen, 2014*). At first glance, this result seems at odds with the attention-induced reduction of variance in the inferred modulators of our model, which generally leads to a decrease in noise correlations. However, a plausible explanation may arise from considering that populations are not just affected by attention-dependent sources of modulation, but also by additional sources that are attention-independent. In the datasets we study here, we identify two examples: the global slow drifts (*Figure 1—figure supplement 1*), and the extra, non-hemispheric, fast modulators (*Figure 2—figure supplement 3*). If the two subpopulations responsible for encoding the two visual stimuli each have their own, separate, attention-dependent modulator, while both populations are subject to an additional set of common, attention-independent modulations, then attention might serve to "unmask" the cross-population correlations arising from these common modulatory signals. Preliminary simulations suggest that such a model could explain these observations.

Finally, the model fits indicate that slow, global drifts and structured, rapid fluctuations in shared gain make substantial contributions to the super-Poisson variability of these neurons. However, the model does not account for all of the observed cue-dependent changes in variability (*Figure 5b*), suggesting that the model structure (e.g. Poisson spiking, with rate modulated by an exponentiated sum of shared signals) is too restricted to capture the full extent of the effects. Moreover, the dimensionality and structure of the estimated modulators are limited by the experiment itself (*Gao and Ganguli, 2015*), and thus cannot be taken as a complete account of modulatory activity in V4. The behavioral task is designed to engage two attentional conditions, each associated with stimuli presented over extensive blocks of trials in one hemifield of the visual world. Similarly, the stimuli in the task provide only a minimal characterization of the selectivity of individual neurons (e.g. receptive field locations, tuning properties). We expect that a more complex task, in which attention is precisely focused in terms of visual location, stimulus properties, and/or time, could potentially reveal modulatory signals targeting more specific subpopulations of neurons.

## Interpreting the model components

Our model is functional, and like other functional models of neural responses, such as those based on Generalized Linear Models (*Truccolo et al., 2005*; *Pillow, 2008*), its components are phenomenological rather than biophysical. The principal value of these models is in providing a parsimonious quantitative framework that explains the relationship between stimuli, neural responses, and behavior. In this respect, a primary contribution of our work is to show that the patterns of neural variability in a visual cortical population during this attentional task are dominated by a few internal signals. This need not have been the case: the model could have required many more modulators to explain the structure of observed population activity, or the imposed structure of the model might have proven inappropriate or insufficient to provide an account of the data.

Notwithstanding the abstraction of the functional model, the inferred properties of the low-dimensional modulatory signal provide constraints on potential underlying mechanisms. We can envision three broad scenarios for the mechanistic source of shared gain fluctuations: bottom-up (stimulus-driven) input, recurrent activity within the population, and top-down state signals. Each of these mechanisms has previously been proposed as an explanation for excess neural response variance or noise correlations (*Goris et al., 2014*; *Ecker et al., 2014*; *Ecker et al., 2012*; *Ecker et al., 2010*; *Zohary et al., 1994*; *Shadlen et al., 1996*; *Shadlen and Newsome, 1998*; *Nienborg and Cumming, 2009*; *Yang and Shadlen, 2007*; *Arieli et al., 1996*; *Litwin-Kumar and Doiron, 2012*). The latter two mechanisms seem compatible with the low-dimensional nature (*Figure 2a*), spatial scale (*Figure 2b–c*), and temporal scale (*Figure 6*) of the modulatory signals we have identified. Top-down and recurrent mechanisms need not operate independently: recent theoretical work proposes a combination of these two mechanisms, in which top-down modulation induces a change in

the balance of excitability in a recurrent E-I network (*Wiese et al., 2014*). One potential top-down biological substrate is cholinergic input, which has a multiplicative (*Disney et al., 2007*) and hemispherically-specific (*Mesulam et al., 1983*) effect on neurons in macaque visual cortex, and is known to be enhanced under attention (*Herrero et al., 2008*). However, it has recently been shown that in area V1, cholinergic input mediates attention-dependent increases in mean firing rate, but not changes in response variance or noise correlations, which appear to depend on NMDA receptors (*Herrero et al., 2013*). Consistent with these findings, our model suggests that the attention-mediated increase and stabilization of gain in area V4 might arise from two separate mechanisms: the estimated coupling weights associated with the cue and modulator signals are similar but not identical (*Figure 5c*), and when they are forced to be identical, model predictions worsen. A full mechanistic account of the modulators could be developed in future experiments by comparing the estimated modulator signals against other neural or physiological signals (e.g. neurotransmitter concentrations, or pupil dilation (*McGinley et al., 2015*; *Reimer et al., 2014*), and by comparing the weights with detailed anatomical and functional properties of the neurons in the population.

## Implications for neural coding

We have used a model that extracts patterns of shared gain variability within a neural population, encapsulating them as "modulators". In turn, we have examined the modulators' statistical structure, and found that it is consistent with a variety of externally measured or controlled quantities. But we are left with an outstanding question: should we think of these modulators as signal or noise? That is, do they reflect a controlled endogenous process, or are they random fluctuations that arise because of a *lack* of control, and are thus detrimental to the encoding of incoming stimuli?

The prevailing view in the neural coding literature is the latter. There has been extensive debate about how correlated fluctuations in neural activity can confound sensory information, and can be difficult to remove once introduced. From this perspective, any process that reduces such fluctuations would improve neural coding (*Cohen and Maunsell, 2009*; *Mitchell et al., 2009*; *Gu et al., 2011*; *Jeanne et al., 2013*; *Downer et al., 2015*; *Zohary et al., 1994*; *Abbott and Dayan, 1999*; *Sompolinsky et al., 2001*; *Wilke and Eurich, 2002*). The benefits of attention would thus be two-fold (*Figure 4*): by increasing the mean gain of a relevant neural population, attention would increase the signal-to-noise ratio (since, for Poisson spike counts, the response standard deviation grows only with the square root of the mean); and by simultaneously reducing the variance of the gain, attention would reduce the overall response variance (*Goris et al., 2014*), and thus the deleterious effects of this nuisance variable.

If this hypothesis is correct, we would expect that gain variability in V4 should follow a general pattern. In the absence of directed attention, all neurons in V4 should be subject to some baseline level of gain variability, due to activity in locally shared circuits. This baseline should presumably be relatively independent of which stimulus features are encoded by which neurons. When attention is directed to a particular neural subpopulation, the shared variability should decrease, improving the coding precision of that subpopulation (*Figure 8a*).

This, however, is not the pattern we observe. The gain fluctuations we uncover do not affect every neuron in the population, but specifically target those neurons relevant to the task (*Figure 8b*). And while their variance certainly decreases under attention, it remains a puzzle why these gain fluctuations are there at all, given that they are nearly absent from the task-irrelevant neurons (which, presumably, would be relevant for different tasks).

This finding suggests that the fluctuating modulatory signals are not noise, but rather reflect meaningful intrinsic signals that play a role in some ongoing computation in the brain. For instance, one tempting possibility is to identify them as fluctuations in the attentional signal itself (e.g. changes in the spatial locus of attention) (*Goris et al., 2014*; *Cohen and Maunsell, 2010*; *Harris and Thiele, 2011*; *Ecker et al., 2012*), or, more generally, as the local manifestation of a dynamic resource allocation strategy. This might reflect a shifting belief-state of the animal in the likelihood or utility of each of the two targets, as proposed by a recent model (*Haefner et al., 2014*). Such an account might explain the modulators' targeting of task-relevant neurons, their low-dimensional structure, and their connection to behavior.

Regardless of what we call this endogenous signal, it is worthwhile considering whether the gain fluctuations it produces pose a problem for downstream interpretation of the neural code. We can envisage three scenarios. First, the downstream neurons that decode the V4 activity may be invariant

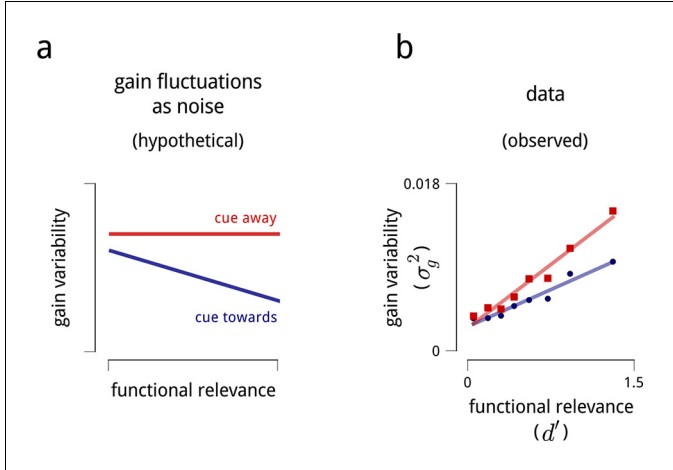

**Figure 8.** Interpreting the role of shared modulation. (**a**) Illustration of how shared gain fluctuations would behave if they were noise, i.e. undesirable random fluctuations. In baseline conditions (red), gain fluctuations would be expected to have similar variance for all neurons in the V4 population. The action of attention would be expected to reduce the variance of gain fluctuations in task-relevant neurons, so as to mitigate their adverse effect on coding precision (see *Figure 4*). (**b**) Contrary to this simple "noise" interpretation, the variance of shared gain fluctuations are markedly larger for task-relevant neurons than task-irrelevant neurons in baseline (cued away) conditions. Moreover, although this variance decreases under attentional cueing (cued toward), it remains larger for the task-relevant neurons. Functional relevance for each unit is measured as $d'$ (as in *Figure 2d–e*); shared gain variability, $\sigma_g^2$, is measured as the total variance of model-estimated gain fluctuations (from slow drift and modulators combined). These results are robust when controlled for firing rate (*Figure 8—figure supplement 1*).

The following figure supplement is available for figure 8:

**Figure supplement 1.** Variance of shared gain fluctuations is larger in task-relevant neurons, even when controlled for firing rate.

to the shared gain fluctuations [for example, in a linear decoding framework, the gain fluctuations might lie parallel to the decision boundaries (*Moreno-Bote et al., 2014*)]. However, this does not appear to be the case: modulator fluctuations have a direct and immediate effect on the perceptual decisions made by the animal (*Figure 7a*).

A second scenario is that downstream neurons might have access to the fluctuating shared gain values, and could thus compensate for their biasing effect on perceptual decisions (e.g. *Stevenson et al., 2010*). This does not imply that the gain fluctuations would have no effect: they would still alter the instantaneous signal-to-noise ratio of the neural representation. This would predict that when a modulator increases, detection probability on the corresponding side would improve (as we see in the top left and bottom right of *Figure 7a*). However, the observation that the value of a modulator also affects performance on the other side (top right and bottom left of *Figure 7a*) is inconsistent with this hypothesis. Any downstream decoder must compare the responses of the two hemispheres in order to perform the task. Thus, the unexpected impact of each hemisphere's modulator on ipsilateral performance demonstrates that the decoder is unable to fully discount the fluctuations in gain.

We are left with a third scenario, in which the downstream neurons cannot perfectly compensate for the shared gain fluctuations in V4. Considered in isolation, shared modulation would thus be detrimental to coding precision. We might speculate that this negative would be offset by some other (as of yet, unknown) benefit of a flexible, varying gain signal.

Overall, we find that the effects of attention on the activity of V4 neurons, while seemingly complex at the level of observed responses, reflect population-level patterns that are simple and low-dimensional. This contributes to an emerging trend in systems neuroscience, wherein the dynamics and statistics of large populations have been found to follow structured, coordinated patterns (*Yatsenko et al., 2015*; *Okun et al., 2015*; *Mazor and Laurent, 2005*; *Broome et al., 2006*;

*Mante et al., 2013*; *Stokes, 2013*; *Kaufman et al., 2014*; *Cunningham and Yu, 2014*). These findings have been driven both by experimental advances in simultaneous acquisition of responses over populations, as well as the development of statistical modeling tools for isolating low-dimensional latent components of population activity (*Kulkarni and Paninski, 2007*; *Paninski et al., 2010*; *Macke, 2011*; *Vidne et al., 2012*; *Archer et al., 2014*). Here, we have shown that these tools may be used to expose and characterize signals that underlie modulatory processes arising under attention.

## Materials and methods

Stimulus design and data processing are described in (*Cohen and Maunsell, 2009*). We only included data from units with mean firing rates greater than 0.5 spikes/s.

### Models

We fitted probabilistic models to population neural responses. For each recording day, we collect the spike counts from the *N* neurons over the *T* stimuli as a (*T* × *N*) matrix, *Y*. We assume that each count was drawn from a Poisson distribution, with the (*T* × *N*) rates obtained from the model response $r_n(t)$ as defined in *Equation (1)*. It is convenient here to express the rates in matrix form, *R*, such that $Y_{t,n} \sim \mathrm{Poiss}(R_{t,n})$. We describe the components and fitting procedures below.

Each neuron *n* has a mean firing rate for the stimulus, $f_n(s_t)$. Since we only analyze responses to the standards, this is a single scalar value per unit (though see section "Fine temporal analysis of shared modulation" below). We define this to be the mean rate in the cue-away condition (with the exception of *Figure 2a*, where the stimulus-drive-only model uses a mean rate across all conditions). This mean rate is modulated by a set of multiplicative factors. Since these factors have to be positive, we adopt the convention that they are derived from signals that can be positive or negative, which are then transformed elementwise by a nonlinear function *h* that is monotonic and has positive range. Choosing *h* as convex and log-concave simplifies inference further (*Paninski, 2004*). The results presented in the main paper use the exponential function exp(·), but we found that using the soft-threshold function log(1 + exp(·)) produces qualitatively consistent results. When using the exponential function, the notation simplifies , and the product of multiplicative factors may be written as a single exponential of a sum, as in Equation (1).

We now describe the gain factors introduced in *Equation (1)*. We define the cue signal during the experiment as a length-*T* binary vector, *c*, indicating the cue direction for each trial. Then we can write the elements of the rate matrix as $R_{t,n} = f_n(s_t) \cdot \exp(c_t\, u_n)$, where $u_n$ is a coupling weight of neuron *n* to the cue signal. Here, it is useful to define the cue signal differently for units in the two hemispheres, with $c_t = 1$ indicating that the cue is directed to a hemisphere's corresponding (contralateral) hemifield, and $c_t = 0$ otherwise. Thus a positive coupling weight $u_n$ means that a neuron increases its gain under cued attention. In matrix notation, we write $C = cu^T$, and $R = F \odot \exp(C)$, with $\odot$ the elementwise product, and *F* as the matrix of values $F_{t,n} = f_n(s_t)$.

Since the joint activity on any day tended to drift slowly over time, independent of the task blocks, we introduced for each recording a slow drift signal, *d*. These were fitted after discounting the cue-dependent gains for each neuron. We constrained each recording's drift signal *d* to vary slowly over time by placing a Gaussian process prior on it. The time constants of the priors were learned for each recording via evidence optimization (*Park and Pillow, 2011*; *Park and Pillow, 2013*), and were typically on the order of minutes to tens of minutes (*Figure 1—figure supplement 1*). The method is described in detail in (*Rabinowitz et al., 2015b*). We jointly fitted the slow drift signal *d*, and the coupling weights of each neuron to this signal, *v*, via an expectation-maximization algorithm. Arranging the effects of these slow, global drifts into a (*T* × *N*) matrix $D = dv^T$, this model amounts to $R = F \odot \exp(C + D)$.

We introduced fast, shared modulators by fitting a (*T* × *N*) gain matrix *M*. The rank of this matrix is a measure of the number of shared modulators: we constrain rank (*M*) = *K* for *K* = 1,2,3,…. In this way, *M* encapsulates the net effect of *K* time-varying modulators $m^{(k)}$, with coupling weights $w^{(k)}$, such that $M_{t,n} = \sum_{k=1}^{K} m_t^{(k)} w_n^{(k)}$. The matrix *M* thus expresses a time-varying latent state of the system (*Kulkarni and Paninski, 2007*; *Paninski et al., 2010*; *Macke, 2011*; *Vidne et al., 2012*; *Archer et al., 2014*). We include a zero-mean Gaussian prior on the elements of *M*, with $p(M) \propto$

$-\frac{\tau}{2}\|M\|_F^2$ to prevent overfitting. We choose the hyperparameter τ for each population and rank by cross-validation. We find the MAP estimate of *M* from the model with $R = F \odot \exp(C + D + M)$, under the low-rank constraint, after fitting *C* and *D* above. Since the low-rank constraint limits the feasible region for the inference problem to a non-convex set, we maximized the log posterior using the Alternating Direction Method of Multipliers (ADMM), in a manner similar to (*Pfau et al., 2013*). We then iterate between solving for *D* and for *C* and *M* (including their respective hyperparameters), though in practice, one or two passes are sufficient for convergence.

We cross-validated model fits as follows. For each stimulus presentation, a random subset of 20% of the spike count observations were set aside as a test set. Thus, during the inference on the training data, the values of the latent variables (in *D* and *M*) for the omitted observations did not contribute to the log likelihood. However, these values were automatically imputed by the priors/constraints on the structure of the latent signals (the drifts *d* to vary slowly in time; the shared modulators *M* to be low rank). After convergence on the training set, the imputed values of these signals were used as predictions on the test set.

## Identifiability

The procedure above recovers a matrix, *M*, of the shared modulators' effects of neural gain, but it is not identifiable: the results do not uniquely constrain the modulator time series (*m*) and the weights (*w*). We resolve these ambiguities as follows.

The rank-1 case arises when fitting separate modulators to each V4 hemisphere (*Figures 2d–e*, *3–5*, *6a–b*, *7*). Here there is a scale and sign ambiguity: we can write $M = m w^{\mathsf{T}} = (\alpha m)\left(\frac{1}{\alpha} w^{\mathsf{T}}\right)$ for any scalar $\alpha \neq 0$. We therefore fix Var(*m*) = 1, and resolve the scale factor into the weights. Since almost all weights for a given hemisphere had the same polarity, we resolve the sign ambiguity so that the mean weight is positive.

The rank-2 case arises when fitting two modulators to a whole population (*Figure 2b–c* only). In addition to the scale/sign ambiguities, there is also a rotation ambiguity: we can write $M = N W^{\mathsf{T}} = (NQ)(Q^{\mathsf{T}} W^{\mathsf{T}})$ for any orthogonal matrix *Q*. For each dataset we choose the rotation *Q* such that the mean weight on the LHS units is $(\bar{w}, 0)^{\mathsf{T}}$ for some value $\bar{w}$. This aligns the brown squares shown in *Figure 2b* and brown points in *Figure 2c* along the x-axis. We resolve the reflection (sign) ambiguity by requiring that the mean weight across both populations lies in the upper right quadrant. This resolution preserves angles, and thus has no bearing on the observed orthogonality of the weight distributions across paired hemispheres shown in *Figure 2b–c* (see *Figure 2—figure supplement 2*).

Our analyses of higher-dimensional models proceed without resolving these ambiguities (*Figure 2—figure supplements 2–3*).

## Behavioral modeling

To quantify the relationship between modulators and behavior, we aggregate the data across all days, and fit a number of generalized linear models (GLMs) to this ensemble.

We estimated how correct detections for cued targets depend on the modulators (*Figure 7a*) by fitting a psychometric curve, with the cued and opposite modulators as regressors. We parameterized the hit probability on trial *t* as:

$$\text{hit}^{(t)} \mid \text{cued target} \sim \text{Bern}\left(\delta \cdot \sigma\left(\alpha + \beta^T m^{(t)} + \lambda \log(\Delta\theta^{(t)})\right)\right)$$

where the superscript (*t*) indicates trial *t*, *m* is a vector of the cued and opposite modulators (averaged across the standard stimuli on that trial), σ is the logistic inverse-link function, Δθ is the target orientation change, δ is a lapse parameter, and the Greek characters α (bias), β (dependence on the modulators), and λ (dependence on task difficulty) are free parameters. Other parameterizations of the dependence on target orientation did not change the main result, nor did omission of the lapse term.

We estimated how correct detections for the opposite targets depend on the modulators by:

$$\text{hit}^{(t)} \mid \text{opposite target} \sim \text{Bern}\left(\sigma(\alpha' + \beta'^T m^{(t)})\right)$$

The values reported in *Figure 7* reflect the average change in hit probability from a unit increase in $m_{cued}$ or $m_{opp}$ via these two models. We also quantify these effects individually for each cued-target condition in *Figure 7—figure supplement 1*; this figure also shows the full psychometrics.

We estimated how correct detections for the cued and opposite targets depend on previous rewards (*Figure 7b*) by replacing the regressors $m^{(t)}$ in the above equations with categorical variables for previous reward (hit for target on cued side / hit for target on opposite side / other), as below.

We estimated how previous reward affects the modulator values (*Figure 7c*) by fitting a Gaussian-GLM:

$$m^{(t)} \sim \mathcal{N}(a + Br^{(t-1)} + Cm^{(t-1)}, \sigma^2 I)$$

where $r^{(t-1)}$ is the reward from the previous trial, being $(1,0)^T$ for a previous hit on a cued target, $(0,1)^T$ for a previous hit on an opposite target, and $(0,0)^T$ for any other outcome (miss/catch trial/false alarm/invalid trial), and $a$, $B$ and $C$ are free parameters.

In all cases, we assessed the significance of parameter estimates by approximating the posterior (both through a Laplace approximation and MCMC) and estimating its integral above/below zero.

## Fine temporal analysis of shared modulation

With the exception of *Figure 6c*, all analyses in the main text assume that the shared modulators are constant in value (and thus have uniform effects) over the course of the response to each stimulus presentation. In *Figure 6c*, we sought to quantify the dynamics of the shared modulator at finer time scales.

We extended the population response model presented in *Figure 1* and *Equation (1)* by allowing the value of the modulator to change over the course of a stimulus presentation. As the data are very limited at fine temporal resolutions, we could not reasonably estimate the modulators' values in small time bins for every stimulus presentation. Instead, we assumed that the temporal evolution of the modulator within each stimulus presentation followed some stereotyped pattern (up to a scale factor that could change from one stimulus presentation to the next).

We extend the response model to capture spike counts within bins of shorter duration (here, 10 ms). We assume that the spike count $Y_{t,b,n}$ for neuron $n$ within bin $b$ of stimulus presentation $t$ is given by:

$$Y_{t,b,n} \sim \text{Poiss}\left(F_{b,n} \cdot exp(C_{t,n} + D_{t,n} + M_{t,b,n})\right)$$

We thus assume that each neuron has a stimulus-driven mean firing rate that changes from bin to bin ($F_{b,n}$), but is identical across repeated stimulus presentations; the cue-dependent gains ($C_{t,n}$) and slow global drift ($D_{t,n}$) are constant over all bins within each stimulus presentation; and the shared modulators ($M_{t,b,n}$) are now free to have structure across stimulus presentations ($t$), neurons ($n$), and bins within each stimulus presentation ($b$). This model is now extremely high-dimensional (the tensor $M$ having $TNB$ free parameters). To overcome this, we must impose structure on $M$, which we do by assuming that it has low tensor rank.

We model the tensor $M$ as an outer product of a rank-2 matrix (with components $M_{t,n}$, as in the two-modulator model presented in the remainder of the text), and a vector $\omega$ indexed over bins, i.e. $M_{t,n,b} = \left(\sum_{k=1}^{2} m_t^{(k)} w_n^{(k)}\right) \omega_b$. The weight vector $\omega$ thus represents the common temporal evolution of the modulators' effects on neural gain within each stimulus presentation, which is identical (up to scale factors) across neurons and successive stimulus presentations.

We learn the whole low-rank tensor of modulator values, $M$, by maximizing the data likelihood. We perform coordinate descent on its components: we iterate between solving for the matrix with elements $M_{t,n}$ (via ADMM), then solving for the vector $\omega$ (via gradient descent). We restrict the fit to the main response period (60 ms to 260 ms after stimulus onset) as the shared fluctuations in spontaneous activity were typically large; the time series $\omega$ is thus shown in *Figure 6c* limited to this period.

## Additional information

### Funding

| Funder | Grant reference number | Author |
|---|---|---|
| Howard Hughes Medical Institute | | Neil C Rabinowitz<br>Robbe L Goris<br>Eero P Simoncelli |
| National Institutes of Health | 4R00EY020844-03 | Marlene Cohen |
| National Institutes of Health | R01 EY022930 | Marlene Cohen |
| Klingenstein Third Generation Foundation | | Marlene Cohen |
| Simons Foundation | | Marlene Cohen |
| Alfred P. Sloan Foundation | Research Fellowship | Marlene Cohen |

The funders had no role in study design, data collection and interpretation, or the decision to submit the work for publication.

### Author contributions

NCR, Conception and design, Analysis and interpretation of data, Drafting or revising the article; RLG, EPS, Conception and design, Drafting or revising the article; MC, Acquisition of data, Drafting or revising the article

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
