## [Decision Letter]

Thank you for submitting your work entitled "Attention stabilizes the shared gain of V4 populations" for peer review at *eLife*. Your submission has been favorably evaluated by Timothy Behrens (Senior editor) and three reviewers, one of whom, Matteo Carandini, is a member of our Board of Reviewing Editors.

The reviewers have discussed the reviews with one another and the Reviewing editor has drafted this decision to help you prepare a revised submission.

Summary:

This paper analyzes the factors that control the activity of populations of neurons, and the impact on those factors of attention and behavior. The paper reanalyzes an impressive data set of population responses (>100 neurons) measured by Cohen and Maunsell (2009, 2010 and 2011) in area V4 of monkeys that were performing an attention task.

One of the paper's achievements is to provide a compact account of the activity of large populations of neurons, and of how this activity is affected by attention. This account is based on a simple model of multiplicative shared variability (similar to those by Goris et al., Lin et al., Ecker et al.). By applying this model, the authors show that "paying attention" to one group of neurons corresponds to increasing the average gain of those neurons, while stabilizing (decreasing the variability across time of) that shared gain.

The model assumes a number of gain modulators and the authors determine the fluctuations of these modulators as a function of attentional cueing and a function of the animal's behavior (and vice versa, how past behavior affects the modulators). The model identifies up to 4 time-varying shared modulatory signals, which affect the neuronal responses of the recorded neurons. These modulators are also linked to past behavior (reward) and are predictive of upcoming performance.

This simple model explains observations that previously appeared contradictory. One is the well-known fact that attention increases firing rates. Another is the fact that attention reduces the correlation between pairs of neurons (Cohen and Maunsell 2009, Mitchell et al. 2009). The two are apparently contradictory, because increases in firing rate gain, on their own, would have caused the opposite effect on correlations. Yet the model captures both: the first by an increase in mean gain, and the second by a decrease in the variance of the gain (which also explains the well-known reduction in variability caused by attention).

The model allows one to assess trial-by-trial variability in a more useful way than simply measuring single neuron variability or pairwise correlations. Moreover, it is a considerable improvement over prior methods to summarize the effect of attention on a large population. The prior methods involved finding a vector between the mean uncued response and the mean cued response and projecting the data onto that axis (Cohen and Maunsell, 2010). The new approach is more general, and finds the directions in neural state space that best account for trial-to-trial shared neural variability with certain broad features. In this way, variability can be assigned to dimensions that have some interpretation, and the contributions of each dimension can be quantified.

Moreover, the paper presents some key new results. Prior methods reported two dimensions of attentional fluctuations, but couldn't quantify how much of the total variability was accounted for by those two dimensions. Here we get a clear and important answer: nearly all of it. Another key new result is that the variability of the attentional fluctuations depends on the attentional cue, being lower in the cued hemisphere than the uncued hemisphere.

The paper also goes some way in clarifying whether the variability in neural activity associated with attention tasks corresponds to 'representation fluctuations' or to 'attentional fluctuations'. In most previous studies, the measured correlations were interpreted as fluctuations in the neural representation of the stimulus. Because correlations often impair the ability to accurately decode, these 'representation fluctuations' were hypothesized to be harmful to the ability to detect changes in the stimulus. Thus, attention would aid detection by increasing signal (by increasing firing rate) and by decreasing noise in the stimulus representation (visible in noise correlations). Cohen and Maunsell 2010 and 2011 described a different form of correlated neural variability: fluctuations in the attentional signal itself. Even when attentional demands are nominally the same across trials, the internal deployment of attention fluctuates, leading to fluctuations in performance: better performance when 'attentional fluctuations' happen to be higher.

Critically, the previous analyses conjured a complex scenario, in which attentional fluctuations would somehow affect representational fluctuations, so that when the attention signal happens to be higher, the representational fluctuations happen to be lower. The present work, instead, indicates a much simpler solution: the decreases in correlated variability caused by attention can be attributed entirely to decreases in the variability of the attentional signals themselves. There is no need to assume attentional modulation of representational fluctuations. Indeed, it isn't even clear if there are large representational fluctuations. Which means we are left with one parsimonious interpretation of the data, which provides a unifying explanation for prior results.

The analysis is well motivated. The paper is very accessible despite its relatively computational approach. It is of interest to a broad audience and is topical.

Essential revisions:

However, the paper also comes with weaknesses, such as the fact that many of its results recapitulate existing results. For example, much of Figure 2 can be inferred from Cohen and Maunsell 2010, including the impact of the cue and two modulatory signals (panel a) and the hemispheric separation (panels b, c). One of the more compelling links to behavior (Figure 6) is essentially a reproduction of the effect in Cohen and Maunsell.

Therefore, for this paper to have impact, it is particularly important to ensure that it does 'provide a unified account of attentional effects on population coding and behavior' (to quote its own words). Yet, the paper seems to fall short of a full commitment to its own findings. Or rather, it commits to the aspects that agree with prior work – there is modulation in dimensions that seem attentional, as in Cohen and Maunsell 2010 – while downplaying the aspects that disagree with prior work. Indeed, a conclusion that can be drawn from the paper is that there is no longer any reason to assume that 'representational fluctuations' are reduced by attention. Rather, this old explanation should be replaced with the new explanation where it is attentional variability itself that is the measured effect. In parts of the Discussion the manuscript commits to that. Yet in other parts, it reverts to explanations of the older variety. For instance, when describing the effect of cholinergic input it reverts to the old model, in which attention causes a reduction in variability. The subsequent paragraph also seems to adhere to the old view. The manuscript seems to avoid undermining prior work, rather than committing to the new unifying interpretation that in other places (including the Abstract and Introduction) it seems willing to endorse.

Ideally, this issue would be tackled not only by changing the Discussion, but by having the model make a new prediction, and look for validation of this prediction in the data. For instance, consider the following idea (provided by one of the reviewers). If the commonly observed decline in 'noise correlations' is in fact due to a decline in attentional fluctuations (rather than a cue-induced suppression of representational fluctuations) then that makes the following prediction. If the data were divided into 'high attention' trials and 'low attention' trials on the basis of the internal signal relevant to that hemisphere, then the gain of the neural response should be higher in the first case, but there should be no difference in the noise correlations. This is a testable prediction.

Another area where the paper should be improved is in explaining which effects are "explained" by the model and which ones are implied as a hypothesis. For instance, in the Discussion, the paper reports that neurons that showed stronger reward dependent changes in firing rates (gain) were also more strongly coupled to the modulator. This is similar to statements made earlier in relation to attentional modulation. This argument may be circular: given the fitting and the associated weights, it may be an inevitable consequence of the computational approach, and thus not a real result. The same applies to the argument in the subsection “Interpreting the model components”, and as stated to wherever similar arguments are made before. These are potentially major criticisms and they absolutely have to be addressed.

Finally, a major comment involves the analysis in Figure 6. This analysis employs the difference between *m_c_* and *m_u_*. In fact, this seems like exactly the wrong thing to do in the current study, as it is shown that those two signals are not correlated. Shouldn't each hemisphere be analyzed on its own?

[Editors' note: further revisions were requested prior to acceptance, as described below.]

Thank you for resubmitting your work entitled "Attention stabilizes the shared gain of V4 populations" for further consideration at *eLife*. Your revised article has been favorably evaluated by Timothy Behrens (Senior editor), by Matteo Carandini (Reviewing editor), and by one reviewer.

The paper has improved considerably and is arguably ready to be published as is. It does a much clearer job at delivering a 'unifying account' of prior results, in the light of new analyses. Where the prior version hedged its bets, the new version takes a clear stand and paints a compelling picture. However, it would be even stronger if the authors would address two main comments indicated below.

The first major comment is about something that is visible in the new version of Figure 7, which shows the impact on behavioral performance of the inferred modulators – one for the cued side and one for the opposite side. This is a better choice than in the previous submission, where they had expressed it as a function of the difference in these two modulators. However, this new analysis reveals something intriguing. The value of the modulator on the cued side predicts the ability to detect a change on the opposite side, even though it does not impact firing rates on the opposite side (top-right red quadrant of Figure 7).

This is remarkable, and seems to completely undercut the theory that the impact of attention on performance is linked to changes in response gain. For example, decreasing the cued-side modulator seems to increase performance on the opposite side, even though this would not increase firing rates in the opposite side, nor decrease their variance. This is because the two modulators are essentially uncorrelated (Figure 6). If this reasoning is correct, the cued-side modulator has an impact on performance for opposite-side stimuli without having any detectable impact on the physiology of the opposite V4 neurons. Thus, the impact of the modulator basically can't be due to physiological effects at the level of V4.

If this understanding of the data is incorrect, it would be a good idea to dispel it. If instead it is correct, it seems that it should be explicitly discussed. The manuscript stops short of this: it mentions that things cannot be fully explained by a "1D axis of attention", and the issue comes up again briefly in Discussion, but it is never really highlighted. Indeed, the finding seems inconsistent with standard models of attention, including the one presented in the paper. And it wouldn't be just about 2D vs. 1D: it would be about whether the physiological effects of attention in V4 have anything to do with the behavioral effects.

The second major comment is about the control analysis currently shown in the reply to the reviewers: dividing the data based on the value of the modulator (high 'attention' versus low 'attention') and assessing whether variability (Fano factor or noise correlations) is lower for the former (which it would be if attention suppresses other variability such as 'representational fluctuations' as envisioned in the old view). It may be a good idea to leave this figure out of the paper, as the results are essentially a foregone conclusion if one accepts the results and interpretation of the prior figures (i.e. if one accepts that the modulator captures essentially all the relevant shared variability). However, some readers might still have doubts about whether the modulators truly account for all the shared variability (that conclusion depends on how suitable the model is) and/or may not immediately see the implications of that finding. It may therefore be a good idea to add that analysis to the supplementary figures.

---

## [Author Response]

Essential revisions:

However, the paper also comes with weaknesses, such as the fact that many of its results recapitulate existing results. For example, much of Figure 2 can be inferred from Cohen and Maunsell 2010, including the impact of the cue and two modulatory signals (panel a) and the hemispheric separation (panels b, c). One of the more compelling links to behavior (Figure 6) is essentially a reproduction of the effect in Cohen and Maunsell.Therefore, for this paper to have impact, it is particularly important to ensure that it does 'provide a unified account of attentional effects on population coding and behavior' (to quote its own words). Yet, the paper seems to fall short of a full commitment to its own findings. Or rather, it commits to the aspects that agree with prior work – there is modulation in dimensions that seem attentional, as in Cohen and Maunsell 2010 – while downplaying the aspects that disagree with prior work. Indeed, a conclusion that can be drawn from the paper is that there is no longer any reason to assume that 'representational fluctuations' are reduced by attention. Rather, this old explanation should be replaced with the new explanation where it is attentional variability itself that is the measured effect. In parts of the Discussion the manuscript commits to that. Yet in other parts, it reverts to explanations of the older variety. For instance, when describing the effect of cholinergic input it reverts to the old model, in which attention causes a reduction in variability. The subsequent paragraph also seems to adhere to the old view. The manuscript seems to avoid undermining prior work, rather than committing to the new unifying interpretation that in other places (including the Abstract and Introduction) it seems willing to endorse.

We agree that we were a bit “restrained” in our interpretations, and appreciate the encouragement. We spent a long time considering your revision comments (as well as the more general comments above), and our discussions led us to a more precise set of interpretations of the results. Although our conclusions are not exactly in the same language as you have described here, we believe they are in the same spirit, and hope you will find them a significant improvement over the original manuscript.

In particular, we were inspired to undertake an additional analysis of the modulator statistics, shown in the new Figure 8, which we think provides strong support for our interpretation that the modulators reflect attentional fluctuations (i.e., what we describe as “signal”), as opposed to representational fluctuations (i.e. what we describe as “noise”). We have substantially revised the Discussion to address this (in particular, replacing the old section “Perceptual benefits of attention” with the new section “Modulators and neural coding”).

We think of the paragraph on cholinergic input (and the subsequent paragraph) as orthogonal to the interpretation of the modulators. We have tried throughout the text to be agnostic regarding the normative role of fluctuating gain signals, and only address this question in the final section of the Discussion. If we have inadvertently given a different impression elsewhere in the text, please let us know so we can correct it.

As an additional side comment, we appreciate that there are similarities between the structures we uncover (e.g. Figure 2) and the structures that Cohen and Maunsell, 2010, assume. In this respect, we emphasise that our goal is to discover the patterns of shared modulation from the population data, rather than assume them. The model produces a set of results – some of which have been previously reported, and some of which are new – but all of them arise from the same unified population model. As per the reviewers’ suggestion below, we have also extended our behavioral results (Figure 7) to a 2dimensional analysis, which further distinguishes our results from those previously published. This also emphasizes the interpretation we land on, that the modulators are signal rather than noise.

Ideally, this issue would be tackled not only by changing the Discussion, but by having the model make a new prediction, and look for validation of this prediction in the data. For instance, consider the following idea (provided by one of the reviewers). If the commonly observed decline in 'noise correlations' is in fact due to a decline in attentional fluctuations (rather than a cue-induced suppression of representational fluctuations) then that makes the following prediction. If the data were divided into 'high attention' trials and 'low attention' trials on the basis of the internal signal relevant to that hemisphere, then the gain of the neural response should be higher in the first case, but there should be no difference in the noise correlations. This is a testable prediction.

This is an interesting suggestion. At a technical level, the idea is correct, and holds up. We can artificially subdivide the trials (within either cue condition) depending on the value of the modulator. We thus create two subsets of trials with different means. If we make sure that the variance of the modulators is the same within the two groups, the noise correlations are very similar (albeit smaller than in the full, combined group). The results of this analysis are included in Figure 9). This validates our argument that the variance of the modulator is a major determinant of the noise correlations.

Author response image 1.Suggested Analysis.Here, we show the results of the reviewers’ suggested analysis. For each day and hemisphere, and within a given condition (cue towards, or cue away), we subdivide the data into two based on the values of the modulator (high modulator values, or low modulator values). We choose the threshold for the split so the variance of the modulator is the same for the two conditions. Top row: the mean firing rates are higher when conditioned on high modulator values. Bottom row: the noise correlations in the two subgroups are roughly equal (though still overall lower in the cue towards condition). Note that the subgroups’ noise correlations are lower than the full dataset for that cue condition, as we have reduced the (conditional) variance in the shared modulator. While this validates our statement that the noise correlations are the result of the variance of the shared modulator (since artificially reducing the variance of the shared modulator causes a dramatic reduction in the noise correlations), it does not resolve the status of whether the modulator is signal or noise.**DOI:**
http://dx.doi.org/10.7554/eLife.08998.020

At a scientific level, however, we don’t believe this provides a strong test of whether the modulators themselves are signal or noise (i.e., whether they capture attentional fluctuations or representational fluctuations), which is the question of interest. Both these hypotheses would provide a complete and selfconsistent description of what we would be doing when we subdivide the trials. However, we were inspired by the spirit of this suggestion to come up with a far stronger test of the two hypotheses, which we present in the new Figure 8.

Another area where the paper should be improved is in explaining which effects are "explained" by the model and which ones are implied as a hypothesis. For instance, in the Discussion, the paper reports that neurons that showed stronger reward dependent changes in firing rates (gain) were also more strongly coupled to the modulator. This is similar to statements made earlier in relation to attentional modulation. This argument may be circular: given the fitting and the associated weights, it may be an inevitable consequence of the computational approach, and thus not a real result. The same applies to the argument in the subsection “Interpreting the model components”, and as stated to wherever similar arguments are made before. These are potentially major criticisms and they absolutely have to be addressed.

Thank you for pointing this out. We agree that there were places where we did not properly distinguish assumptions from findings. We’ve made an attempt to address this throughout the manuscript. With regard to the specific example about the rewardcoupling of individual neurons, we decided that this was a relatively minor point, and unnecessarily confusing. As such, we’ve removed this point, and feel that the manuscript is better off without it. Please let us know if there are any remaining concerns.

*Finally, a major comment involves the analysis in Figure 6. This analysis employs the difference between* m_c_
*and* m_u_*. In fact, this seems like exactly the wrong thing to do in the current study, as it is shown that those two signals are not correlated. Shouldn't each hemisphere be analyzed on its own?*

Thanks for raising this point. The difference does capture much of the effect, but we went back and reanalyzed the two signals separately, and have now included this in the paper. We also simplified this figure (now Figure 7) as it provided unnecessary detail to the understanding of the major result (this detail is now in Figure 7—figure supplement 1).

[Editors' note: further revisions were requested prior to acceptance, as described below.]

The paper has improved considerably and is arguably ready to be published as is. It does a much clearer job at delivering a 'unifying account' of prior results, in the light of new analyses. Where the prior version hedged its bets, the new version takes a clear stand and paints a compelling picture. However, it would be even stronger if the authors would address two main comments indicated below.

The first major comment is about something that is visible in the new version of Figure 7, which shows the impact on behavioral performance of the inferred modulators – one for the cued side and one for the opposite side. This is a better choice than in the previous submission, where they had expressed it as a function of the difference in these two modulators. However, this new analysis reveals something intriguing. The value of the modulator on the cued side predicts the ability to detect a change on the opposite side, even though it does not impact firing rates on the opposite side (top-right red quadrant of Figure 7).This is remarkable, and seems to completely undercut the theory that the impact of attention on performance is linked to changes in response gain. For example, decreasing the cued-side modulator seems to increase performance on the opposite side, even though this would not increase firing rates in the opposite side, nor decrease their variance. This is because the two modulators are essentially uncorrelated (Figure 6). If this reasoning is correct, the cued-side modulator has an impact on performance for opposite-side stimuli without having any detectable impact on the physiology of the opposite V4 neurons. Thus, the impact of the modulator basically can't be due to physiological effects at the level of V4.If this understanding of the data is incorrect, it would be a good idea to dispel it. If instead it is correct, it seems that it should be explicitly discussed. The manuscript stops short of this: it mentions that things cannot be fully explained by a "1D axis of attention", and the issue comes up again briefly in Discussion, but it is never really highlighted. Indeed, the finding seems inconsistent with standard models of attention, including the one presented in the paper. And it wouldn't be just about 2D vs. 1D: it would be about whether the physiological effects of attention in V4 have anything to do with the behavioral effects.

We think this interpretation is incorrect, and we had provided an explanation of this in the last section of the Results in the manuscript. The structure of the task is such that the animal must compare the represented stimuli in both hemispheres: it is a threealternative task (left target, right target, or no target). Thus, even though the modulators are uncorrelated, the value of a modulator on one side can affect performance on the other.

We appreciate that the explanation is a bit subtle, and it was easy to miss in the Results. To make this point more salient in the manuscript, we have added a few sentences to the Discussion (see end of paragraph beginning with “A second scenario…”).

The second major comment is about the control analysis currently shown in the reply to the reviewers: dividing the data based on the value of the modulator (high 'attention' versus low 'attention') and assessing whether variability (Fano factor or noise correlations) is lower for the former (which it would be if attention suppresses other variability such as 'representational fluctuations' as envisioned in the old view). It may be a good idea to leave this figure out of the paper, as the results are essentially a foregone conclusion if one accepts the results and interpretation of the prior figures (i.e., if one accepts that the modulator captures essentially all the relevant shared variability). However, some readers might still have doubts about whether the modulators truly account for all the shared variability (that conclusion depends on how suitable the model is) and/or may not immediately see the implications of that finding. It may therefore be a good idea to add that analysis to the supplementary figures.

Thank you for this question. This prompted us to realize that we have somehow failed to show, in a quantitative way, exactly how well the model accounts for the attentioninduced changes in variability (i.e. the Fano factor and the noise correlation). The plot in Figure 2 does indicate the overall performance of the model in capturing the responses, but this is in units of log likelihood (which are not easily interpretable). Figure 5 demonstrates that we can predict reductions in variability, and that the magnitude of these effects depends on the strength of coupling to the hidden modulator. But what is missing is a quantitative statement of how well the model predicts the reductions in Fano factor and noise correlation.

Given this, we have generated yet another figure (Figure 5), and we hope you will find that it satisfies your request. We think this is a more direct demonstration than the figure we sent with our previous responses, and we have included it in the main text.